# Enhancing Fish Auction with Deep Learning and Computer Vision: Automated Caliber and Species Classification

Javier Jareño [1], Guillermo Bárcena-González [1,*], Jairo Castro-Gutiérrez [2,3], Remedios Cabrera-Castro [2,4] and Pedro L. Galindo [1]

1 Computer Science Department, School of Engineering, University of Cádiz, Puerto Real, 11519 Cádiz, Spain; javier.jareno@uca.es (J.J.); pedro.galindo@uca.es (P.L.G.)
2 Biology Department, Faculty of Marine and Environmental Science, Campus Universitario de Puerto Real, University of Cádiz, Puerto Real, 11510 Cádiz, Spain; jairo.castro@uca.es (J.C.-G.); reme.cabrera@uca.es (R.C.-C.)
3 Department of Agroforestry Sciences, ETSI School of Engineering, University of Huelva, 21007 Huelva, Spain
4 Instituto Universitario de Investigación Marina (INMAR), Campus de Excelencia Internacional del Mar (CEIMAR), University of Cádiz, Puerto Real, 11510 Cádiz, Spain
* Correspondence: guillermo.barcena@uca.es

**Abstract:** The accurate labeling of species and size of specimens plays a pivotal role in fish auctions conducted at fishing ports. These labels, among other relevant information, serve as determinants of the objectivity of the auction preparation process, underscoring the indispensable nature of a reliable labeling system. Historically, this task has relied on manual processes, rendering it vulnerable to subjective interpretations by the involved personnel, therefore compromising the value of the merchandise. Consequently, the digitization and implementation of an automated labeling system are proposed as a viable solution to this ongoing challenge. This study presents an automatic system for labeling species and size, leveraging pre-trained convolutional neural networks. Specifically, the performance of VGG16, EfficientNetV2L, Xception, and ResNet152V2 networks is thoroughly examined, incorporating data augmentation techniques and fine-tuning strategies. The experimental findings demonstrate that for species classification, the EfficientNetV2L network excels as the most proficient model, achieving an average F-Score of 0.932 in its automatic mode and an average F-Score of 0.976 in its semi-automatic mode. Concerning size classification, a semi-automatic model is introduced, where the Xception network emerges as the superior model, achieving an average F-Score of 0.949.

**Keywords:** fish species; fish size; machine learning; deep learning; transfer learning; fish auction; classification

## 1. Introduction

The quest for intelligence in the Spanish fish market has become imperative to enhance and modernize the existing infrastructure. While most auction facilities have digitized their operations, the current level of digitalization remains limited, primarily concentrating on photographing products for potential buyers. However, there is room for improvement in this regard. Expanding the use of these images, particularly in the realm of biological research, holds significant promise. Therefore, enhancing digital infrastructure can open doors to broader applications, including in the field of biology, such as technological advancements in sales traceability [1] and auction processes [2] to augment the value of fishing along the Spanish coast. In the specific case of the fish markets in Conil de la Frontera (36°17′44.1″ N 6°08′16.9″ W) and Sanlúcar de Barrameda (36°48′12.7″ N 6°20′12.3″ W) in Southwest Spain, an image capture and tagging system is used exclusively to maintain sales records and for online sales.

During the auction, the seller presents a box of fish of the same species and size, which is indicated by an integer value [1, n], where $n > 1, n \in \mathbb{N}$ depending on the species

(smaller integer indicating larger size). The size of the lot is determined by the physical characteristics of the catch, which are verified by the auction manager. Once both labels are manually confirmed, the lot is displayed to potential buyers and offered for auction.

However, despite efforts to ensure objectivity in the labeling process, this task has relied on manual processes, rendering it vulnerable to subjective interpretations by the involved personnel. Hence, mislabelling may occur. Instances include concealing smaller fish within larger ones, combining physically similar but differently valued species, mislabeling catches with smaller sizes to increase their value, and variations in size depending on the smallest and largest catch of the day. These cases highlight the necessity of implementing an auxiliary classification system that enables the auction operator to manage sales with the utmost rigor and accuracy.

The utilization of convolutional neural networks (CNNs) combined with data augmentation techniques is a well-established approach to similar tasks, widely acknowledged in the scientific community. Previous studies [3–8] that focus on the classification of animal species have demonstrated the effectiveness of convolutional models. They propose species classification systems based on pre-trained CNNs such as ResNet-50 [9], VGG16 [10], AlexNet [11], and GoogleNet [12], utilizing datasets augmented with techniques like rotation, zoom, and shift. Furthermore, these works highlight the improved performance achieved by combining convolutional models with preprocessing techniques, data augmentation, and fine-tuning during training.

This study investigates the implementation of an automated system for species and size recognition in fish auctions using convolutional neural networks (CNNs). The main difference between automatic and semi-automatic models in classification lies in the degree of human intervention required for their operation and supervision. In this work, we explore multiple pre-trained models and examine the impact of various image preprocessing techniques on the performance of an automatic classification system. Additionally, we propose a semi-automatic model wherein the auction manager is presented with a reduced set of predicted species based on their probabilities, and the final species label will be chosen.

In most existing publications, specimens are only classified within the image or distinguished from other objects in the image, such as trash [6,13–17]. Our research addresses the critical need for objectivity in fish labeling processes, which are traditionally prone to subjective errors. This system not only promises to streamline auction operations by reducing manual errors but also significantly contributes to technological advancement in the field of biological research and fishery management.

The structure of this paper is as follows: Section 2 presents the images dataset along with the data augmentation strategy used to handle class imbalance and increase the robustness of the system to new instances (Section 2.1). The evaluated pre-trained models are discussed in Section 2.2, followed by the selection of hyperparameters for training (Section 2.3) and the evaluation of the proposed models (Section 2.4). Subsequently, the results of the proposed models are discussed in Section 3. Finally, the conclusions and future work of this study are presented in Section 4.

## 2. Materials and Methods

### 2.1. Image Acquisition and Preprocessing

The data set used was collected from sales conducted at the Conil de la Frontera fish market. Each sold box is associated with an image (800 × 480 px resolution), which is stored along with auction sales data, including size, weight, and Food and Agriculture Organization of the United Nations Code (FAO), among other data pertaining to the private data of both buyers and sellers. Upon examining the captured images of fish in our study, the overlap of various specimens in several instances is evident. This overlap is not accidental, nor does it represent a limitation of our methodology. On the contrary, it accurately reflects the natural and realistic conditions under which the fish capture images have been taken. Example images of the two most significant species in the Conil de la Frontera fishing port are shown in Figure 1. The original raw dataset comprises 12,525 images spanning 80 different species

of sales occurring on 38 distinct days. However, a notable class imbalance is observed, with several species having fewer than 30 sample images, while more common species such as those depicted in Figure 1a,b have 1217 and 2167 instances, respectively.

To address the class imbalance issue and ensure an adequate test dataset, we decided to study those species for which we have more than 200 sales instances. Consequently, the set of target species is reduced to 19. This set is divided into training (80% of the data for each species), validation (10%), and testing (10%). The dataset is divided in a manner such that every species has 80% of their instances in training, 10% for validation, and 10% for testing, henceforth keeping the balance in all the stages.

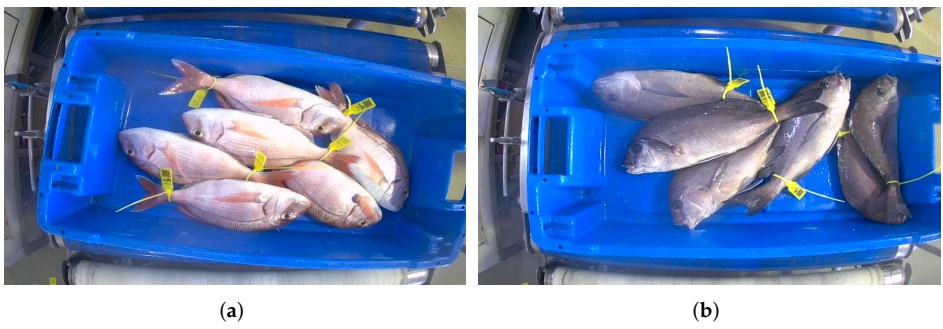

(**a**)            (**b**)

**Figure 1.** Main species of the fish market at the fishing port of Conil de la Frontera: (**a**) *Pagrus pagrus* and (**b**) *Plectorhinchus mediterraneus*.

The 19 species selected according to the criterion are listed in Table 1, resulting in a total of 10,632 instances. Despite spanning 38 different days and 19 species, all these images share identical characteristics: intense artificial lighting within a controlled environment, ensuring no disruption to the day-night cycle during image capture; variable fish counts per box, allowing for significant overlap; consistent camera and configuration settings across all shots; each box containing fish of the same species; and use of the same kind of box (disposable white boxes are used for species with ink, although these instances were not considered due to insufficient data).

Data augmentation techniques are applied to the training data [18], with the aim of increasing the number of instances for each species to at least 500. The implementation of data augmentation significantly enhances the system's performance, as it allows the network to learn from variations present in the images, such as the distribution of fish in the box, different calibers, blood stains, water droplets on the camera distorting the image, and snow in the boxes. This augmentation process facilitates the extrapolation of the network's knowledge to future instances. The original set of images is preserved, and new images are generated using transformations (*Albumentations framework* [19]) such as horizontal mirroring (0.5 probability), vertical mirroring (0.5 probability), image rotation (range from −10º to 10º in steps of 1º), blur (maximum Gaussian kernel size for blurring the input image set as 1), optical distortion (distort limit 0.05), and hue saturation (hue shift limit of 5, value shift limit of 5 and saturation limit in range [0, 30]). Therefore, the resulting dataset consists of 10,632 instances, representing 19 target species, with an average of 640 images per class. Furthermore, 20% of the dataset is reserved for test and validation purposes. Therefore, it is divided into training (80%—8505 original instances, 12,095 with data augmentation), validation (10%—1064 instances), and testing (10%—1063 instances).

**Table 1.** Number of instances recorded for each species, alongside their respective scientific names and identification codes.

| Specie | ID | No. of Instances |
|---|---|---|
| *Conger conger* | 1203 | 292 |
| *Pagellus erythrinus* | 1501 | 889 |
| *Plectorhinchus mediterraneus* | 1504 | 2167 |
| *Argyrosomus regius* | 1506 | 836 |
| *Pagrus auriga* | 1509 | 1120 |
| *Sparus aurata* | 1510 | 261 |
| *Pagrus pagrus* | 1515 | 318 |
| *Dentex gibbosus* | 1519 | 212 |
| *Diplodus sargus sargus* | 1520 | 313 |
| *Dentex canariensis* | 1524 | 761 |
| *Diplodus vulgaris* | 1528 | 208 |
| *Umbrina canariensis* | 1606 | 287 |
| *Microchirus azevia* | 1607 | 233 |
| *Phycis phycis* | 1704 | 380 |
| *Pagrus pagrus* | 1705 | 1217 |
| *Muraena helena* | 1803 | 265 |
| *Galeorhinus galeus* | 1802 | 241 |
| *Mustelus mustelus* | 1804 | 335 |
| *Diplodus cervinus cervinus* | 1911 | 297 |

## 2.2. Pre-Trained CNN models

A convolutional neural network is a type of deep learning model specifically designed to process grid-structured data, such as images or matrix data. Unlike traditional neural networks, CNNs use a specialized architecture that takes advantage of the spatial correlation of data by applying convolutional filters in successive layers [20]. These filters automatically learn local features, such as edges, textures, and shapes, and combine them to build more abstract and meaningful representations as you dig deeper into the network.

One of the main advantages of convolutional neural networks is their ability to extract features automatically from the input data without prior preprocessing. Instead of having to design and manually select the relevant features for a particular task, CNNs automatically and hierarchically learn the most discriminative features as the network is trained. This makes them extremely powerful for computer vision tasks such as image classification, object detection, segmentation, and face recognition.

Pre-trained networks are CNN models that have been pre-trained on large datasets, such as ImageNet, and have become a popular choice in the deep learning community [21]. These pre-trained models, such as ResNet152V2 [22], VGG16 [10], EfficientNetV2L [23], and Xception [24], have learned to recognize a wide variety of visual features and thus can be used as a solid foundation for other computer vision tasks. By leveraging the prior knowledge captured in these models, significant time and resources can be saved by avoiding training from scratch and obtaining quality results more quickly.

Let the VGG16 CNN architecture shown in Figure 2 work as an example. It cannot be used without applying transfer learning [20], a technique used to extrapolate pre-trained CNN models to new classification problems. CNNs can be divided into two main parts: the convolution layers, where the feature extraction is performed, and the fully connected layers, where the classification is performed based on the features extracted. Since the original number of classes is different from ours, the last part of the CNN architecture is replaced with a new Fully Connected Neural Network (FCNN), which matches our number of fish species, 19.

The original feature extraction layers are maintained since they hold the knowledge learned from the ImageNet dataset. Notwithstanding, the classification layers are replaced with a *GlobalAveragePooling2D* layer and a *Dense-SoftMax* output layer.

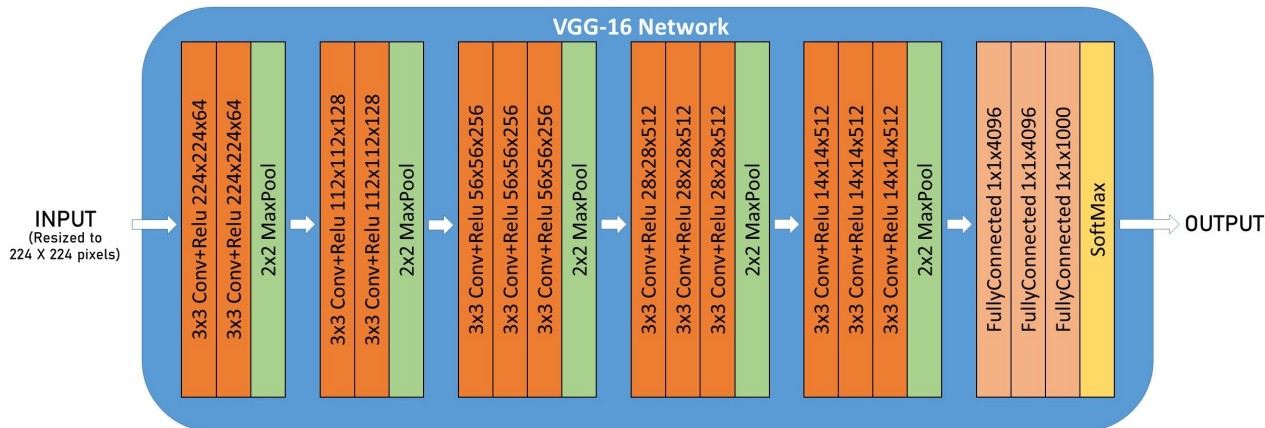

**Figure 2.** VGG16 CNN architecture.

*2.3. Model Training*

Since the transfer learning technique has been used, the model training is performed in two phases to improve the performance of the results. First, the convolutional layers of the model are frozen so that the training will only affect the fully connected (FC) layers. We use "categorical cross-entropy" as the loss function and the Adam optimizer with its default hyperparameters of learning rate and decay [20]. The model is trained in 35 epochs and with a batch size of 64. These parameters have been chosen to properly fit the model in the shared memory of the GPU to optimize training time. The GPU is an NVIDIA GeForce RTX 3090Ti with 24 GiB of memory, of which 22.4 GiB is used for model storage.

Once the model has been fully trained with these parameters, a fine-tuning training phase is performed to adapt the whole network to this specific problem and increase its performance. In this stage, all the layers are unfrozen so that the training changes all the weights of the model. However, the learning rate is set as $1 \times 10^{-5}$. so that the weights are not changed that much. Moreover, because we are updating all the layers, the size of the model in the GPU is increased. Therefore, the batch size must be reduced to 16. All these parameters are shown in Table 2.

**Table 2.** Summary of Training Hyperparameters: The batch size has been determined based on the GPU's memory size limitation. The selection epoch values for both training phases are selected after an analysis of the learning curves, as depicted in Figure 3, which exhibits signs of overfitting at those particular points. During the fine-tuning phase, an increase in the number of epochs significantly impacts the validation error.

|  | Optimizer | Loss Function | Learning Rate | Epochs | Batch Size |
|---|---|---|---|---|---|
| Frozen Layers | Adam | Categorical Cross-entropy | $1 \times 10^{-3}$ | 35 | 64 |
| Fine-Tuning |  |  | $1 \times 10^{-5}$ | 5 | 16 |

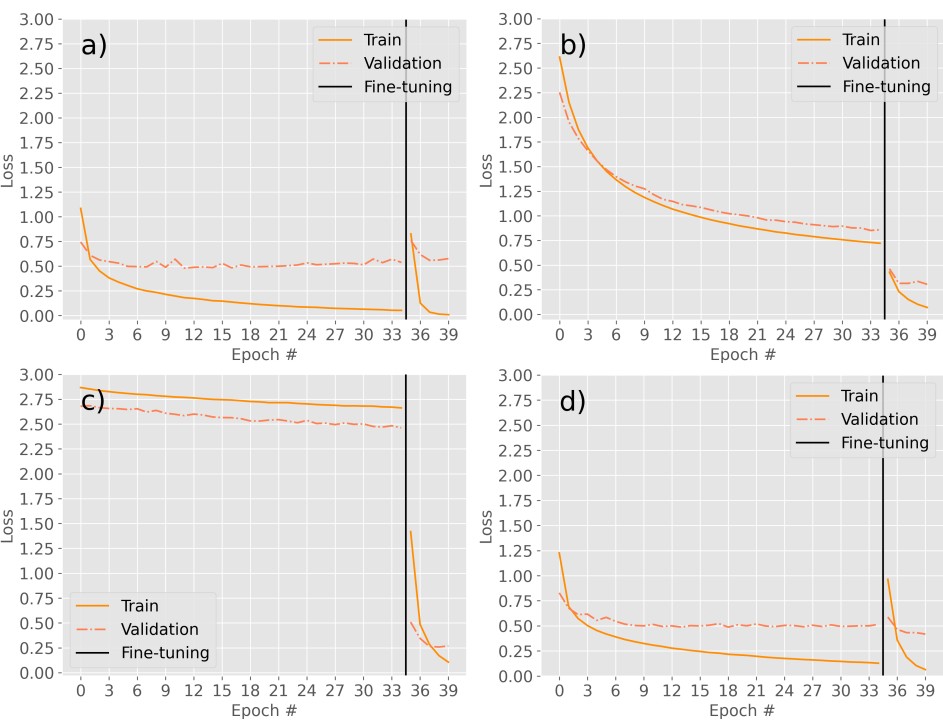

**Figure 3.** Training and validation errors during both phases of training. (**a**) ResNet152V2, learning curve shows the quick fit to 0.5 error value. (**b**) VGG16, slow progressive learning curve, training is prematurely stopped due to the strong fit in its fine-tuning phase. (**c**) EfficientNetV2L, slow learning during the first phase, great improvement in fine-tuning. (**d**) Xception, a quick fit of the validation error in both phases.

### 2.4. Model Evaluation

To evaluate the model performance, we delve into the concept of model evaluation specifically tailored for CNN models. The metrics commonly used to assess the effectiveness of CNNs in various computer vision tasks will be explored. These metrics offer valuable insights into the model's ability to classify objects. Examining and understanding these evaluation metrics allows for informed decisions regarding model selection, optimization, and deployment.

The following equations define the metrics employed to measure the performance of the models, which are Precision, Recall, $F_1$ Score, Execution Time, and Confusion Matrix.

$$\text{Precision} = \frac{TP}{TP + FP} \tag{1}$$

$$\text{Recall} = \frac{TP}{TP + FN} \tag{2}$$

$$F_1\text{Score} = 2 \cdot \frac{\text{precision} \cdot \text{recall}}{\text{precision} + \text{recall}} \tag{3}$$

where $TP$ stands for True Positives, $FP$ for False Positives and $FN$ as False Negatives. The confusion matrix allows us to visualize the predicted classes among the real ones, giving an intuitive way to understand the previous metrics and to see the most common confusion between the classes.

A semi-automatic model is proposed so that the second most likely class predicted (second highest value in the one-hot encoding) is taken as a $TP$, too. This metric is used to evaluate the model that would be proposed as a product for the fish markets. This model proposes the most likely predicted class for the fish box but also will show the next $n$ most

likely species in case the predicted species is wrong. Therefore, this metric evaluates the final model.

To ensure a robust evaluation of the proposed models, a total of 50 distinct datasets were meticulously curated. Instead of adhering to the commonly used minimum of 30 iterations, we opted for this approach to account for the considerable variations in the number of classes and differences among images. This approach allows us to encompass a wide range of scenarios representing sets of instances that can be input into the network. Hence, with respect to each of the experiments, the metrics presented are the result of 50 iterations of each model proposed.

## 3. Results and Discussion

This section presents the results from a series of experiments conducted on multiple pre-trained models. The study investigates the performance of VGG16, EfficientNetV2L, ResNet152V2, and Xception, each evaluated independently. To ensure the quality and accuracy of the analysis, the networks underwent training iterations using diverse training datasets derived from the original set. This approach enables a comprehensive comparison among the models under various training conditions.

### 3.1. Species Classification

The learning curves of the proposed models across the epochs show how the training stops once the validation loss has achieved a stable value and the training loss is beginning to overfit (Figure 3). Moreover, the same loss function has been used over the models, and although they report similar performance in the F-Score metric (as seen in Table 3), they obtain very different values in their loss functions.

**Table 3.** F-Score, Precision, and Recall evaluation metrics are presented for the classification of 19 species among 50 different datasets. EfficientNetV2L consistently demonstrates superior performance across all evaluation metrics. The reported mean values of precision and recall align consistently across various statistical measures, encompassing minimum, maximum, and mean values. Consequently, the F-Score serves as a reliable and representative indicator of model performance.

| | | VGG16 | EfficientNetV2L | Xception | ResNet152V2 |
|---|---|---|---|---|---|
| F-Score | Min | 0.894 | 0.911 | 0.862 | 0.824 |
| | Max | 0.946 | 0.955 | 0.908 | 0.877 |
| | Mean | 0.918 | 0.932 | 0.886 | 0.853 |
| | Std ($\sigma$) | 0.011 | 0.008 | 0.01 | 0.011 |
| Precision | Min | 0.899 | 0.91 | 0.867 | 0.826 |
| | Max | 0.947 | 0.957 | 0.908 | 0.878 |
| | Mean | 0.923 | 0.935 | 0.888 | 0.855 |
| | Std ($\sigma$) | 0.01 | 0.008 | 0.01 | 0.011 |
| Recall | Min | 0.894 | 0.913 | 0.862 | 0.828 |
| | Max | 0.946 | 0.955 | 0.908 | 0.876 |
| | Mean | 0.919 | 0.932 | 0.886 | 0.854 |
| | Std ($\sigma$) | 0.011 | 0.008 | 0.01 | 0.012 |

The metrics obtained from various models are presented in Table 3. The experimental results highlight the exceptional performance of the EfficientNetV2L network, surpassing all other models. Although VGG16 produces similar metrics, EfficientNetV2L consistently achieves superior results across all evaluation metrics. Conversely, both Xception and ResNet152V2 demonstrate inferior performance compared to the aforementioned models, as their metric values fall below the minimum values obtained by EfficientNetV2L.

Furthermore, it is noteworthy that the precision and recall mean values reported by each model coincide across all statistical measures, including minimum, maximum, and

mean. Thus, the F-Score serves as an accurate representation of the proposed model's performance. Illustratively, Figure 4 depicts the confusion matrix derived from one evaluation iteration of the EfficientNetV2L network, attaining an impressive F-Score of 0.943. The matrix effectively exhibits the accurate classification of nearly all instances, which is evident from the dominant values along the main diagonal.

Nevertheless, it becomes evident that three instances pose considerable challenges in terms of classification. These instances correspond to species identified as 1515 (Dentón—*Dentex gibbosus*), 1524 (Vieja—*D. canariensis*), and 1705 (*P. pagrus*). These closely related species share numerous characteristics, including color, shape, fins, and size. Consequently, accurate image classification becomes exceedingly difficult if the distinctive features distinguishing these species are not clearly discernible in the photographs. For instance, a key distinguishing characteristic between *P. pagrus* and *D. canariensis* lies in the presence of a red spot on the tail fin of *P. pagrus* (see Figure 5).

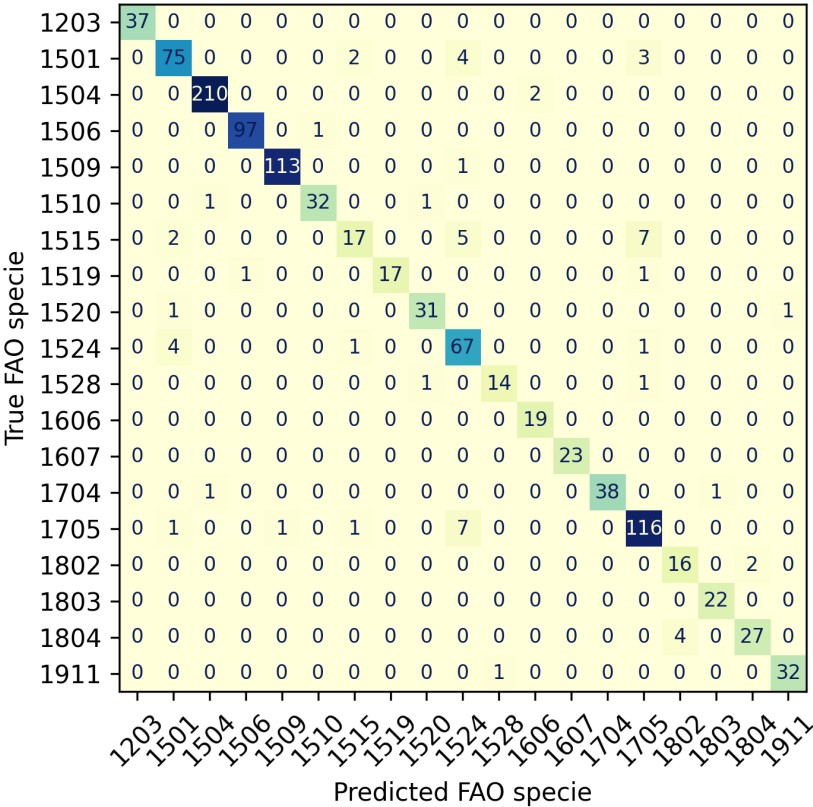

**Figure 4.** EfficientNetV2L Confusion Matrix reporting a F-Score 0.943, close to the mean values (shown in Table 3). A distinct diagonal pattern can be seen, highlighting the model's well-performing nature. The primary areas of confusion for the model are observed in the comparisons between 1515–1524, 1515–1705, 1705–1524 (True Specie—Predicted Specie) comparisons.

The histogram depicted in Figure 6 illustrates the distribution of values obtained for the F-Score metric. This histogram is presented due to its significance as the most representative measure for evaluating the performance of the model. Notably, the histograms show a Gaussian-like distribution pattern, indicating that the mean value can serve as a statistically significant indicator of the model's performance. Furthermore, this observation is reinforced by the examination of the mean and standard deviation values, as outlined in Table 3, wherein the standard deviation consistently hovers around the range of 0.01.

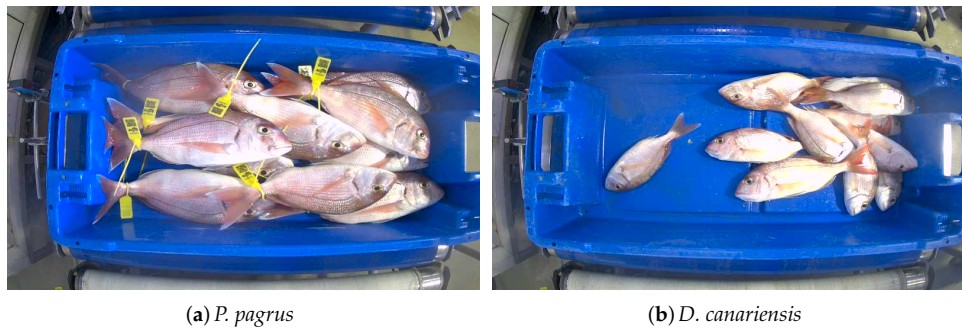

(**a**) *P. pagrus*                    (**b**) *D. canariensis*

**Figure 5.** Difference between *P. pagrus* (**a**) and *D. canariensis* (**b**). Notice the difference in the intensity of the red color between the fins of both species, with the *P. pagrus* being much more saturated than that of the *D. canariensis*.

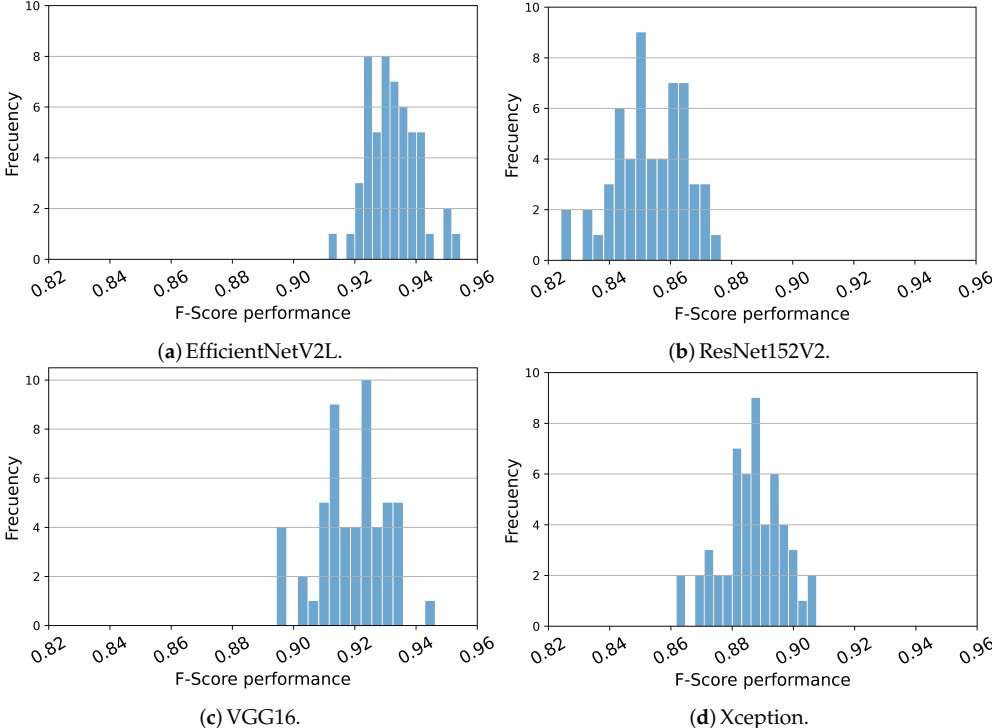

(**a**) EfficientNetV2L.                    (**b**) ResNet152V2.

(**c**) VGG16.                    (**d**) Xception.

**Figure 6.** Histograms of F-Score for the proposed models among 50 different datasets. (**a**) Efficient-NetV2L, (**b**) ResNet152V2, (**c**) VGG16 and (**d**) Xception. The histograms closely align with their respective mean and standard deviation values (seen in Table 3), therefore serving as significant indicators of the mean value. In panel (**a**), it is evident that the mean F-Score values reported for this model surpass most executions of other models, approaching the performance level achieved by the model depicted in panel (**c**).

Histograms of F-Score for the proposed models are presented in these figures. These histograms closely align with their respective mean and standard deviation values, therefore serving as significant indicators of the mean value. In panel (a), it is evident that the mean F-Score values reported for this model surpass most executions of other models, approaching the performance level achieved by the model depicted in panel (c).

F-Score, Precision, and Recall evaluation metrics are presented for the classification of 19 species among 50 different datasets. EfficientNetV2L consistently demonstrates superior performance across all evaluation metrics. The reported mean values of precision and recall align consistently across various statistical measures, encompassing minimum, maximum, and mean values. Consequently, the F-Score serves as a reliable and representative indicator of model performance.

The study encompassed the analysis of the semi-automatic version of the models, with the corresponding results shown in Table 4. A careful examination of the table reveals that EfficientNetV2L consistently outperforms other models in terms of performance. Notably, VGG16 now demonstrates values that closely approach the performance of EfficientNetV2L. Furthermore, Xception exhibits a noteworthy improvement in results, approaching the performance levels of the previously mentioned models. However, despite delivering acceptable performance, ResNet152V2 does not exhibit notable improvement compared to the other models.

**Table 4.** F-Score, Precision, and Recall evaluation metrics are presented for the semi-automatic classification of 19 species among 50 different datasets. EfficientNetV2L consistently demonstrates superior performance across all evaluation metrics, while VGG16 exhibits a notable performance improvement that brings it closely in line with the performance of EfficientNetV2L. Consequently, both models effectively serve as suitable systems for the semi-automatic classification of species.

|  |  | **VGG16** | **EfficientNetV2L** | **Xception** | **ResNet152V2** |
|---|---|---|---|---|---|
| F-Score | Min | 0.961 | 0.966 | 0.941 | 0.922 |
|  | Max | 0.983 | 0.986 | 0.972 | 0.957 |
|  | Mean | 0.972 | 0.976 | 0.956 | 0.937 |
|  | Std ($\sigma$) | 0.006 | 0.004 | 0.007 | 0.007 |
| Precision | Min | 0.962 | 0.967 | 0.942 | 0.923 |
|  | Max | 0.983 | 0.986 | 0.972 | 0.959 |
|  | Mean | 0.973 | 0.977 | 0.957 | 0.938 |
|  | Std ($\sigma$) | 0.005 | 0.004 | 0.007 | 0.008 |
| Recall | Min | 0.96 | 0.966 | 0.941 | 0.924 |
|  | Max | 0.983 | 0.986 | 0.972 | 0.957 |
|  | Mean | 0.972 | 0.976 | 0.956 | 0.937 |
|  | Std ($\sigma$) | 0.006 | 0.003 | 0.007 | 0.007 |

Of particular interest is the remarkable average F-Score value achieved by EfficientNetV2L, coupled with its low standard deviation. This observation further solidifies the effectiveness of the semi-automatic version of the model. Moreover, an examination of the histogram depicting the F-Score values for this particular model, as shown in Figure 7, illustrates a narrower distribution. Consequently, this indicates a more reliable and consistent model suitable for deployment in the fishing port setting.

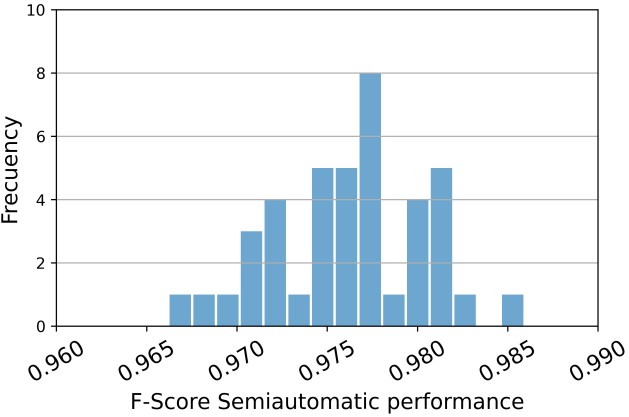

**Figure 7.** Histogram of EfficientNetV2L semi-automatic model's F-Score among 50 different datasets. The minimal deviation and consistently high F-Score values underscore its reliability, substantiating its suitability for implementation within the auction market.

### 3.2. Size Classification

In these experiments, 50 runs of various models were executed, yielding a historical record of metric outcomes. The species size classification under consideration is that of *P. pagrus*, which can assume 5 distinct calibers, where 1 corresponds to the largest and 5 to the smallest. There are a total of 1217 instances, which are partitioned into training (10%), validation (10%), and test (10%) sets.

The ensuing results pertain to a semi-automatic model, wherein predictions falling within a maximum error range of $\pm 1$ in comparison to the actual size are considered True Positives (TP), in extreme cases, 1 is considered valid along with 2, and in case 5, 4 is considered valid alongside 5. This choice is attributed to the high subjectivity inherent in the original data annotations. Fishermen, when presenting their catch for sale, propose an initial caliber for the specimen, which is then verified and annotated by the auction supervisor, either confirming the original size or making corrections as necessary. The criteria employed in this process are subjective and contingent upon the number and size of the daily catches, causing variations in the visual reference system. Consequently, specimens weighing 1 kg by law may belong to class 3, yet on certain days, they are categorized as class 2. Moreover, it is not deemed practical to manually review and amend the data according to an objective criterion, as it does not align with the protocol employed at the auction house and thus holds no relevance therein. Hence, the approach of permitting a maximum error of 1 is adopted, therefore affording the operator the option to either accept the system's suggestion or modify it by 1.

The metrics obtained from the various models are compiled in Table 5. Within this table, it is evident that Xception emerges as the most reliable model across all metrics, slightly surpassing the results achieved by ResNet152V2 and significantly outperforming VGG16 and EfficientNetV2L. The low standard deviation value indicates that the model has demonstrated robustness across multiple executions with diverse instances from the database. This behavior is reflected in all three proposed metrics.

**Table 5.** F-Score, Precision, and Recall evaluation metrics are presented for the semi-automatic classification of size among 50 runs. EfficientNetV2L and ResNet152V2 consistently exhibit superior performance across all evaluation metrics, with Xception being the model that surpasses all other models in terms of metrics.

|  |  | **VGG16** | **EfficientNetV2L** | **Xception** | **ResNet152V2** |
|---|---|---|---|---|---|
| F-Score | Min | 0.852 | 0.547 | 0.924 | 0.902 |
|  | Max | 0.948 | 0.818 | 0.983 | 0.975 |
|  | Mean | 0.901 | 0.702 | 0.949 | 0.942 |
|  | Std ($\sigma$) | 0.025 | 0.059 | 0.017 | 0.023 |
| Precision | Min | 0.9 | 0.49 | 0.926 | 0.906 |
|  | Max | 0.956 | 0.809 | 0.984 | 0.977 |
|  | Mean | 0.926 | 0.714 | 0.952 | 0.945 |
|  | Std ($\sigma$) | 0.014 | 0.075 | 0.016 | 0.021 |
| Recall | Min | 0.876 | 0.661 | 0.926 | 0.901 |
|  | Max | 0.95 | 0.868 | 0.983 | 0.975 |
|  | Mean | 0.914 | 0.774 | 0.949 | 0.942 |
|  | Std ($\sigma$) | 0.02 | 0.044 | 0.016 | 0.023 |

Furthermore, the alignment between the minimum, maximum, and mean values of precision and recall metrics establishes the F-Score as an accurate representation of the proposed model's performance.

For illustrative purposes, Figure 8 presents a confusion matrix from one of the executions of the Xception model, achieving an F-Score of 0.959. It can be observed that the instances of confusion are minimal, rarely exceeding a value of 2. A specific illustrative case

is highlighted in which confusion arises between the smallest and largest size categories. Upon manual verification, it becomes evident that the instances should objectively belong to class 3, and confusion arises due to the proximity of these specimens to the center of the camera frame, where distortion is minimal. The fisheye lens effect in the camera introduces distortion, causing fish located at the edges of the container to appear significantly smaller than those placed in the center.

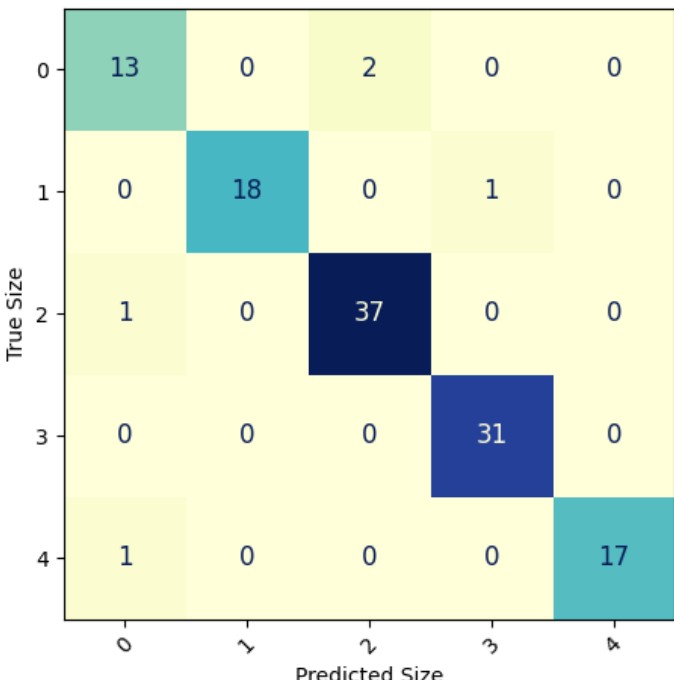

**Figure 8.** Xception Confusion Matrix reporting a F-Score 0.959 close to the mean values (shown at Table 5). The instances of confusion are minimal, rarely exceeding a value of 2. A specific illustrative case is highlighted in which confusion arises between the smallest and largest size categories. Upon manual verification, it becomes evident that the instance should objectively belong to class 3, and confusion arises due to the proximity of these specimens to the center of the camera frame, where distortion makes it bigger than it is.

Finally, to provide detailed insight into the experimentation process, manual verification of classification errors has been conducted, both in terms of size and species. These verifications reveal that the model exhibits classification errors in instances where fish reflect excessive light, in boxes with very few fish positioned at the farthest end of the camera's range (resulting in significant distortion), where there is excessive condensation inside the box (due to cooling systems), when an identifying label obstructs a significant portion of the fish in the box, or when there are foreign objects present. Conversely, it is expected that a change in the image acquisition infrastructure will not significantly affect species classification as long as certain characteristics such as resolution, image quality, distortion, and distance to the box are maintained. However, in the case of size classification, adjustments may be necessary to ensure that the aspect ratio and size of the fish are similar across both cameras, as this factor plays a crucial role in caliber classification.

Nonetheless, it is noteworthy that even when dealing with subjectively annotated data and substantial image distortion, the semi-automatic Xception model stands out as a potential candidate for deployment in sales supervision.

## 4. Conclusions

The experimental results demonstrate the exceptional performance of the Efficient-NetV2L network compared to other models evaluated in the study regarding species classifi-

cation. EfficientNetV2L consistently outperformed VGG16, Xception, and ResNet152V2 in terms of the evaluated metrics, establishing its superiority across all aspects.

The precision and recall mean values reported by each model align consistently across various statistical measures, corroborating the F-Score as an accurate representation of the proposed model's performance. The confusion matrix derived from one evaluation iteration of the EfficientNetV2L network, with an impressive F-Score of 0.943, vividly displays the accurate classification of most instances, as demonstrated by the dominant values along the main diagonal. The histogram analysis of the F-Score metric further supports the evaluation of model performance.

In contrast, the semi-automatic model, which utilizes the pre-trained EfficientNetV2L network and is intended for deployment in fishing ports, has exhibited an average F-Score of 0.98 with a standard deviation of 0.003. These findings substantiate the model's strength and readiness for practical implementation for species classification.

This study has provided compelling evidence of the exceptional performance exhibited by the pre-trained EfficientNetV2L network in comparison to the other proposed networks, encompassing both its automatic and semi-automatic versions. Its consistent and remarkable performance across evaluations confirms its suitability for practical implementation. Moreover, an important issue has been identified regarding the classification of closely related species, necessitating more extensive image-processing techniques for future investigations.

Regarding the study on size classification, it has examined the classification of species size, specifically focusing on *P. pagrus*, which can be categorized into five distinct calibers. The experiments involved multiple model runs, resulting in a comprehensive historical record of metrics. Notably, the analysis employed a semi-automatic approach, where predictions within a maximum error range of $\pm 1$ were considered True Positives due to the inherent subjectivity in the original data annotations, influenced by factors such as varying catch sizes and fisherman-supervisor interactions. Despite this subjectivity and the practical impracticality of manually revising the data, the approach allowed for a maximum error of 1, granting operators the flexibility to accept system suggestions or adjust them accordingly. The evaluation revealed Xception as the most reliable model, surpassing other architectures, with low standard deviation indicating robustness across diverse dataset instances. The alignment of precision and recall metrics demonstrated that the F-Score accurately represented model performance. An illustrative confusion matrix highlighted the model's effectiveness even in cases of distortion, making the semi-automatic Xception model a promising candidate for sales supervision despite subjectively annotated data and image distortions.

As for future work, it is proposed to employ a significantly larger database, enabling the classification of all species handled at the auction house. Regarding the size classification system for a species, the suggestion is to develop a system based on an objective criterion. This would involve counting the number of pieces in a container and calculating the weight of each one, as the total weight of the container is readily available. In this way, it is possible to obtain average weights and sizes of specimens, information that is crucial both for sales and of great biological significance for the species, since with long-term data series, it could be detected whether sizes are decreasing over the years, which can be a clear indication of resource overexploitation. Furthermore, by using the calibration table that governs the objective classification, it would be possible to establish an objective classification system that adheres to the official regulations.

**Author Contributions:** Methodology, J.J.; Software, J.J.; Validation, R.C.-C.; Formal analysis, J.C.-G.; Investigation, G.B.-G.; Resources, J.C.-G.; Writing—original draft, J.J. and G.B.-G.; Writing—review and editing, P.L.G.; Supervision, R.C.-C. and P.L.G. All authors have read and agreed to the published version of the manuscript.

**Funding:** The work was supported by "Ministerio de Agricultura, Pesca y Alimentación—Fondos NextGenerationEU" (DIGIPESCA Project) and "Junta de Andalucía—Grupos PAI" (TIC-145 and RNM-243 Research Groups).

**Institutional Review Board Statement:** Regarding the ethical considerations of our study, it is pertinent to note that the specimens used in this work have never been subjected to animal experimentation. These specimens come from catches made by professional fishermen and are subject to European and National regulations. Therefore, our research did not require Ethics Committee or Institutional Review Board approval, as it involved the analysis of specimens obtained through standard regulatory-compliant practices.

**Informed Consent Statement:** Not applicable.

**Data Availability Statement:** Data is unavailable due to privacy at the time of publishing. However, some efforts are being made to make this data publicly available.

**Acknowledgments:** The authors would like to thank the Lonja de Conil (fish market) for providing the data used in this research.

**Conflicts of Interest:** The authors declare no conflict of interest.

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
