# Peer review of "Enhancing Fish Auction with Deep Learning and Computer Vision: Automated Caliber and Species Classification"

_fishes, doi:10.3390/fishes9040133_

Round 1

Reviewer 1 Report

Comments and Suggestions for Authors

The manuscript entitled 'Enhancing Fish Auction with Deep Learning and Computer Vision: Automated Caliber and Species Classification' employs pre-trained convolutional neural network (CNN) models to automate the classification of species and determination of size class. Specifically, both automated and semi-automated models have been utilized for species classification, while semi-automated models have been applied to size caliber determination.

Firstly, I did not identify any apparent technical flaws, and the authors have provided the requisite information to facilitate the reproducibility of the results. Furthermore, the results are promising, presented in a clear and well-structured manner. However, from my perspective, there is room for improvement in the following major and minor points.

Major points:

  1. Please highlight the images/photos utilized in this research are of high quality throughout the manuscript, exhibiting clarity and a lack of fish overlap. The only potential issue may be the varying proximity of the fish to the camera's center frame.
  2. It is noteworthy to highlight that the purpose of this application is to enhance objectivity in the preparation for auctions, as the current results may not be adequate to serve as determinants of fish quality and pricing.
  3. Permit me to raise a personal inquiry, as I am unfamiliar with the intricacies of fish auctions in Spain. In my limited experience with other auction practices, typically only items of high value (e.g., exceptional freshness, I guess) are deemed suitable for auction, still necessitating manual intervention. If this is the case, I perceive this work as an initial trial into automating auction processes.

Minor point

  1. In the Methods section, it would be beneficial if the authors could reinforce the distinction between the automatic and semi-automatic models employed.
  2. The explanation provided in Lines 201-207 could be more comprehensible if accompanied by relevant visual aids or photos.
  3. In Figure 3, there appears to be a typographical error, where 'phirst' should be corrected to 'first.'
Comments on the Quality of English Language

Minor editing of English language required

Reviewer 2 Report

Comments and Suggestions for Authors

The manuscript presents a study on the application of deep learning for automated caliber and species classification. This is an interesting theme to be explored and the results reported in the manuscript can be useful for the scientific community. However, there are a few weaknesses that need to be properly addressed, as detailed below.
- The characterization of the current state of the art is quite limited. I would not expect a Review Paper level of detail, but more information is certainly needed about aspects such as the techniques that have been used for species recognition, the way they have evolved, and the potential weakness that still need suitable solutions. A proper contextualization is very important to highlight the importance of the work being reported.
- Still in the introduction, the contributions of the study should be more clearly emphasized, using current research gaps as a way to emphasize the importance of the progress made in this study.
- The first paragraph of Section 2 should either be moved to the introduction, or simply be removed.
- Line 68: why the images collected are of such a small resolution? Was this decision deliberate or was this the result of limitations in the image capture process?
- Section 2.1: a Table showing all the species present in the dataset and the number of samples associated to each of them would be very useful. Additionally, it would be useful to have a more detailed description of the database characteristics. For example, all images were captured under the same illumination conditions, or lighting characteristics vary considerably? The number of fish in each box is always the same, or is there some variation? This type of information helps the reader to proper evaluate the difficulty of the classification task.
- Lines 177-178: why were these specific architectures chosen? Was there a pre-selection step in which the models which would most likely produce good results were chosen?
- Figures 4 and 7: the confusion matrices are more useful when percentage values are presented, instead of absolute numbers. An even better option is to present both percentage and absolute values together.
- One of the most difficult problems in the classification of images captured under uncontrolled conditions is the variability that arises from this situation. The discussion part of the manuscript is missing an analysis about the impact of factors such as illumination, background interference, etc. Also, it is not clear if the models would be robust to factors like camera model and image resolution.

Comments on the Quality of English Language

There are some minor problems that can be easily corrected.

Round 2

Reviewer 2 Report

Comments and Suggestions for Authors

I am satisfied with the responses provided by the authors.

Comments on the Quality of English Language

Apart from a few minor issues, language is fine.